# Look the Other Way: Designing 'Positive' Molecules with Negative Data via Task Arithmetic

## Abstract

The scarcity of molecules with desirable properties (i.e., 'positive' molecules) is an inherent bottleneck for generative molecule design. To sidestep such obstacle, here we propose molecular task arithmetic: training a model on diverse and abundant negative examples to learn 'property directions' – without accessing any positively labeled data – and moving models in the opposite property directions to generate positive molecules. When analyzed on 33 design experiments with distinct molecular entities (small molecules, proteins), model architectures, and scales, molecular task arithmetic generated more diverse and successful designs than models trained on positive molecules in general. Moreover, we employed molecular task arithmetic in dual-objective and few-shot design tasks. We find that molecular task arithmetic can consistently increase the diversity of designs while maintaining desirable complex design properties, such as good docking scores to a protein. With its simplicity, data efficiency, and performance, molecular task arithmetic bears the potential to become the *de-facto* transfer learning strategy for de novo molecule design.

## 1 Introduction

Discovering one drug molecule can take over a decade and billions of dollars (Wouters et al., 2020). The first obstacle is charting the 'chemical space' effectively (Bohacek et al., 1996). Chemical space is estimated to contain approximately $10^{60}$ drug-like molecules, with a scarcity of molecules possessing desirable properties, e.g., bioactivity towards a pharmaceutically relevant target. Generative deep learning has emerged as a revolutionary technology for drug discovery – with the potential to shorten de novo design pipelines from years to weeks (Wu et al., 2024; Ghazi Vakili et al., 2025).

Generative drug discovery faces an inherent challenge: molecules with desirable properties (e.g., bioactivity) are not only scarce but also may lack structural diversity. In the case of early-stage drug discovery, positive molecules (or 'hits') can be as rare as 1% in large, diverse molecule libraries. For new pharmacological targets, identifying bioactive molecules is often time-consuming, with the vast majority of candidates proving inactive compared to the few that show activity. Finally, bioactive molecules are often optimized through minor structural edits to explore structure-activity relationships, resulting in medicinal chemistry datasets typically containing a few hundred molecules with low structural diversity (Sun et al., 2017; Tran-Nguyen et al., 2020; van Tilborg et al., 2022). Transfer learning has served to mitigate data scarcity by (a) pretraining on large unlabeled molecular sets, then (b) finetuning on molecules with desirable properties (e.g., bioactivity) (Segler et al., 2018). Despite this, the constraints of drug discovery data – where active compounds are vastly outnumbered by inactive ones, or might not be available – can still limit model effectiveness. In this context, leveraging the large wealth of 'negative' data is an unexplored avenue to boost the potential of generative deep learning for drug discovery. Stemming from this observation, this work introduces task arithmetic for the first time into the molecular sciences, as an effective strategy for chemical space exploration leveraging negative data.

Task arithmetic manipulates model weights to transfer or combine learned properties across tasks (Ilharco et al., 2023). It has shown promise in vision and language applications, such as model merging and multi-objective optimization (Ilharco et al., 2023; Choi et al., 2024). Here, we apply

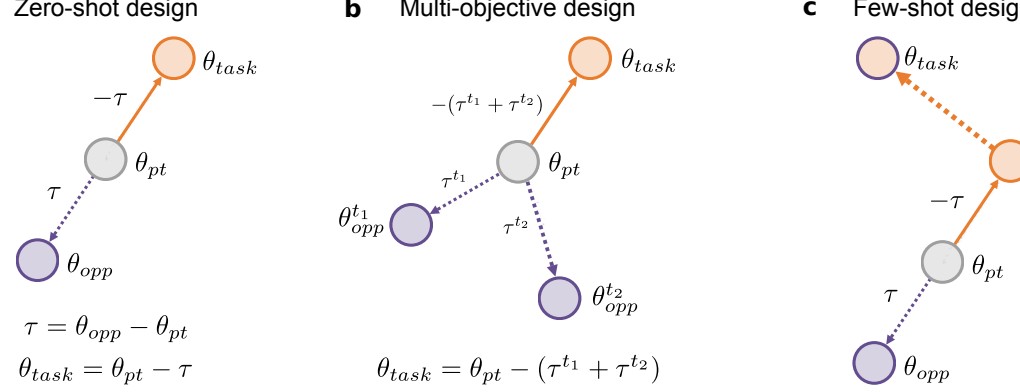

Fig. 1: *Molecular task arithmetic (MTA).* **(a)** Zero-shot design. Molecular task arithmetic learns a task direction in the model weight space by finetuning on negative molecules (dashed purple arrow). The task vector is traversed in the opposite direction (solid orange arrow). **(b)** Multi-objective design. Multiple task directions are learned independently, and then combined. **(c)** Few-shot design. Task arithmetic is applied on the model finetuned with negatives, and then known positive molecules are used (dashed orange arrow).

it to an intriguing challenge: training with negative data to steer models away from undesirable chemical regions in order to generate positive molecules. This is an especially daunting task because (a) chemical space is vast, discrete, and sparse, and (b) structure-property relationships are non-linear. Here, we investigate whether task arithmetic can open new opportunities to efficiently chart the dark chemical universe. Our large-scale and systematic study across 33 (bio)chemical design tasks shows that it can design *positive* molecules by using *negative* ones, thereby remarkably advancing the capabilities of generative deep learning to navigate the chemical space.

The key contributions of this work are the following:

- We introduce *molecular task arithmetic* (MTA) in generative drug design for the first time, as a strategy to overcome data limitations typical of drug discovery and leverage negative molecules in a novel way.

- We show that MTA enables *zero-shot ligand-based de novo* design, by designing molecules fulfilling *one* or *multiple* target properties starting from negative molecules, even in cases where no positive molecules were available to the models.

- We augment *few-shot molecule design* with task arithmetic. When used to augment traditional transfer learning pipelines, MTA increases the number of diverse and desirable designs across tasks of increasing difficulty.

Our results display the potential of molecular task arithmetic to replace dominant ligand-based de novo design practices, by leveraging molecules with *undesirable* properties in an unprecedented way.

## 2 BACKGROUND AND RELATED WORK

**Task arithmetic.** Certain directions in the weight space of a trained model – task vectors – correspond to a change in performance on specific tasks (Ilharco et al., 2023; Yang et al., 2025). In a transfer learning setting, where data for a task $t$ are available, a task vector $\tau^t$ can be computed by subtracting the pretrained model weights from the finetuned model weights (Ilharco et al., 2023):

$$\tau^t = \theta_{\text{ft}}^t - \theta_{\text{pt}}, \tag{1}$$

where $\theta_{\text{ft}}^t$ and $\theta_{\text{pt}}$ are the flattened weights of the finetuned and of the pretrained model, respectively. The task vector can then be used to modify the behavior of the pretrained model, e.g., subtracting the task vector from the pretrained model weights can yield a model that is worse at the finetuning task $t$:

$$\theta_{\text{new}} = \theta_{\text{pt}} - \lambda\tau^t, \tag{2}$$

where $\lambda$ is a scaling factor that determines the step size in the direction. Termed as "forgetting via negation", this approach can remove undesired behavior from deep models, such as toxic language generation (Ilharco et al., 2023). Task vectors can also be added together to create models with multi-task capabilities – "learning via addition" (Ilharco et al., 2023):

$$\theta_{\text{mobj}} = \theta_{\text{pt}} + \lambda(\tau^{t_1} + \tau^{t_2} + \tau^{t_3} + \cdots + \tau^{t_n}), \tag{3}$$

where $\tau^{t_i}$ is the task vector learned for the task $t^i$. In Eq. 2 and 3, $\lambda$ controls how far the final model lands from the pretrained one in the weight space. While too large $\lambda$ values have detrimental side effects on model performance, too low values fail to edit the model behavior sufficiently.

**Molecule design with deep learning.** Early deep learning for molecule design approaches trained sequence models on string representations of molecules with a next-token prediction language modeling task (Olivecrona et al., 2017; Segler et al., 2018). Recent works used molecular graphs and point clouds to represent molecules, combined with graph neural networks and diffusion (Xia et al., 2019; Wang et al., 2025). While these approaches enable more fine-grained representation and higher steerability in learning, decoder-only language models of molecular strings, termed chemical language models, remain popular to date due to their simplicity and performance (Grisoni, 2023). SMILES (Fig. A1a) is the most popular string representation for chemical language modeling. In a transfer learning setting, two phases typically occur: (a) *pretraining*, a sequence model is trained with a language modeling task (Fig. A1b), on several millions of molecules to learn key elements, e.g., generating 'chemically valid' SMILES and basic molecular properties; and (b) *(self-)supervised finetuning*, where a curated set of molecules with desired properties is used, with the same training objective to condition the model (Skinnider et al., 2021). New molecules can then be designed by sampling the learned multinomial distribution autoregressively. This pipeline has become a well-established standard (Merk et al., 2018; Wu et al., 2024; Ghazi Vakili et al., 2025).

## 3 MOLECULAR TASK ARITHMETIC

*Why task arithmetic?* Chemical space is vast but sparse: it is estimated to contain $10^{60}$ molecules, but molecules of desirable properties are rare. Current finetuning strategies rely on these rare, 'positive' molecules and therefore, the limited data availability and diversity form a bottleneck. Here we 'look the other way' and propose a new transfer learning strategy that aims to leverage the abundant and diverse 'negative' molecules, that is, molecular task arithmetic (MTA).

MTA views "forgetting via subtraction" (Eq. 2) as an opportunity to learn *only* from negative data to design positive molecules. For the task $t$ of designing molecules with a desirable property, MTA finetunes a pretrained model ($\theta_{pt}$) with molecules having non-desirable values of this property. This yields a model ($\theta_{opp}$) that can generate negative molecules. The task vector $\tau^t$ is then computed by subtracting $\theta_{pt}$ from $\theta_{opp}$. Finally, a new model ($\theta_{task}$) can be constructed by subtracting $\tau^t$ from $\theta_{pt}$ with a scaling factor $\lambda$ (Fig. 1a):

$$\tau^t = \theta_{\text{opp}} - \theta_{\text{pt}},$$
$$\theta_{\text{task}} = \theta_{\text{pt}} - \lambda\tau^t. \tag{4}$$

Our main scientific question is whether $\theta_{task}$ can generate *chemically-valid* molecules with *desirable* properties. In other words, whether molecular task arithmetic can learn the 'direction' of a molecular property using negative molecules and then 'look the other way' – by moving the pretrained model in the opposite direction to design positive molecules. By not using any labeled positive molecules, if successful, task arithmetic might open doors to unexplored tasks, e.g., zero-shot ligand design.

## 4 EXPERIMENTAL SETTINGS

**Physico-chemical properties.** As a first systematic analysis, we applied MTA to five physico-chemical properties that are relevant for drug-likeness, and at the same time are easy and fast to compute: (a) fraction of sp3-hybridized carbons (frac. sp$^3$C); (b) octanol-water partitioning coefficient (logP); (c) number of hydrogen bond donors; (d) number of rings (No. Rings); and (e) topological surface area (TPSA). We defined conditional molecule design experiments as designing molecules with a higher or lower value than a predefined threshold of the five selected molecular properties

(Fig. A2), for a total of 10 design tasks. For each task, we curated five training, validation and test splits (containing 1024, 256 and 256 positive molecules, respectively). Additionally, we defined three dual-objective tasks: (a) high fraction of $sp^3$-hybridized carbons and a high number of hydrogen bond donors, (b) high lipophilicity and low TPSA, and (c) high number of rings and low TPSA. We selected minimally correlated properties (absolute correlation lower than 0.3; Table A1) to ensure neither conflicting nor trivial tasks.

**Ligand-target docking.** We applied MTA to achieve a more challenging task: designing molecules that dock well into a protein target using negative data. We chose three pharmacologically relevant proteins: (a) coagulation factor II (F2), (b) glucocorticoid receptor (NR3C1), and (c) peroxisome proliferator-activated receptor delta (PPARD). We curated (a) binding molecules with good docking scores for supervised finetuning and (b) non-binding molecules with poor docking scores for molecular task arithmetic (*see* Appendix for details). Both well-docking and poorly-docking molecules constitute less than 5% of the pretraining set, further increasing the difficulty of the task (Fig. A3).

**Overall setup.** We relied on chemical language models (long-short term memory networks, LTSMs, 3.17M parameters) pretrained on 1.5M SMILES strings from ChEMBL (Gaulton et al., 2017). For any task, we (a) finetuned models with positive molecules, and (b) trained molecular task arithmetic models with negative molecules. What follows is structured based on the following objectives:

1. *Generating 'chemically valid' molecules*, where we evaluate if the model weight manipulation introduced by task arithmetic destructs generative capabilities of the model.

2. *Achieving zero-shot de novo design*, where only molecules with 'undesired' properties are used to design 'desirable' molecules, both for single- and dual-objective design (Fig. 1a,b).

3. *Understanding the effect of task vector magnitude*, where we analyze how scaling the molecular task vector via $\lambda$ affects chemical space exploration for desirable molecules.

4. *Boosting few-shot molecule design* (Fig. 1c), by augmenting traditional finetuning (which relies on positive molecules only) with negative molecules, via molecular task arithmetic.

5. *Designing well-docking molecules*, where we study if molecular task arithmetic can harness poor-docking molecules to design well-docking ones, as a proxy for designing bioactive molecules starting from inactive ones.

Moreover, as an additional modality, we applied molecular task arithmetic to *designing highly-ordered proteins*. This serves to study if molecular task arithmetic can be extended to additional molecular entities and model architecture (large-scale pretrained transformer, (Ferruz et al., 2022)) to design proteins with fewer disordered regions, starting from disordered proteins.

## 5 RESULTS

### 5.1 CAN TASK ARITHMETIC EVEN GENERATE VALID MOLECULES?

The language of SMILES strings possesses a strict grammar to represent molecules, where single-character changes might yield strings that cannot be converted back into molecules, i.e., invalid SMILES strings. Hence, our first question was whether the capacity to generate valid SMILES is retained under the model weight manipulation introduced by task arithmetic. To answer this question, we applied molecular task arithmetic (MTA) with $\lambda = 0.50$ (Eq. 4) across the 20 single-objective design tasks, consisting of two SMILES formats. We then computed validity (the ratio of valid and unique SMILES strings among 100,000 generations), as a well-established measure of a model's capacity to learn SMILES syntax. Molecular task arithmetic obtains an average validity of 79.0% with randomized SMILES strings and 59.2% with canonical SMILES strings. While these values drop from the pretrained models (which were optimized for validity, obtaining values of 95.2% and 95.8% for randomized and canonical SMILES strings, respectively), MTA preserves its ability to generate valid molecules, especially with randomized SMILES strings. Remarkably, this finding indicates that the weight space of a chemical language model can be traversed with no loss guidance – opening unexplored avenues in generative deep learning for molecule design.

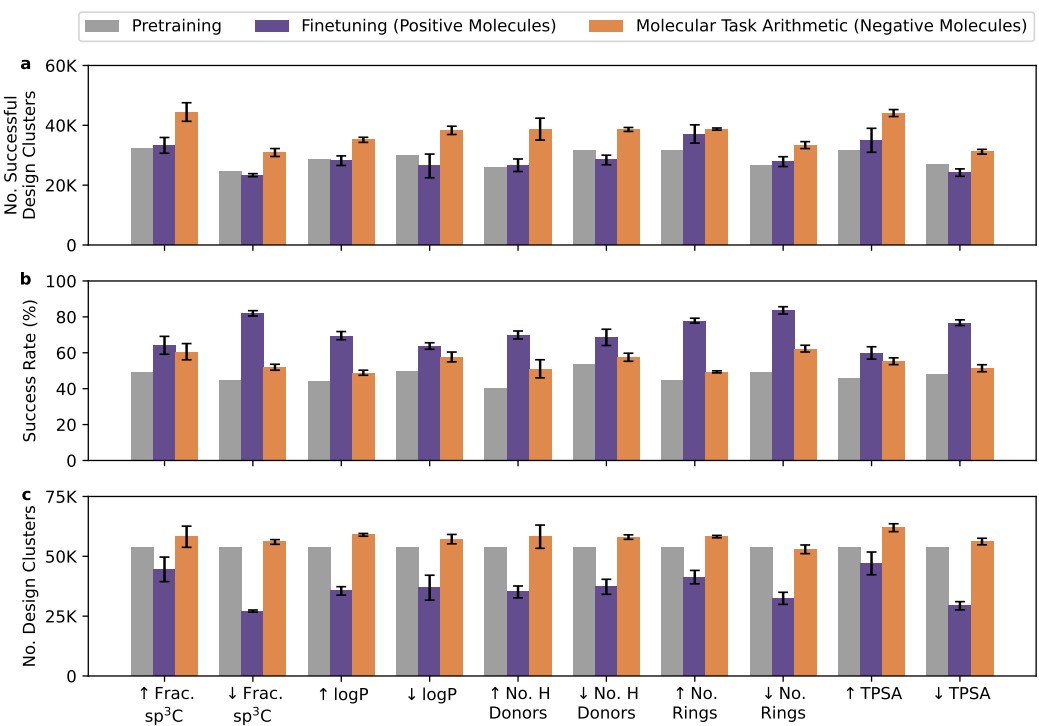

Fig. 2: *Zero-shot single-objective molecule design.* Molecular task arithmetic *vs* fine-tuning, across 10 design tasks (100,000 molecular designs). The pretrained model is included as a baseline, and average and standard deviation are reported. **(a)** number of cluster centers that possess the desired property; **(b)** ratio of designs that satisfy the design task; **(c)** number of clusters.

## 5.2 Zero-Shot Molecule Design

**Single-objective design.** We benchmarked MTA with (a) supervised finetuning and (b) pretraining only. We experimented with varying values of $\lambda$ (controlling the deviation from the pretrained model, Eq. 4), in particular, between 0.10 and 1.00 with a step size of 0.05. We experimented with fractions of the finetuning dataset (composed of 1024 molecules) ranging from 1% to 100%, with increments of 10%. For task arithmetic, 1024 negative molecules (finetuning set of the opposite property) were used. We evaluated the models by the *number of successful design clusters*, i.e., number of structurally-distinct molecular clusters identified by `rdkit`'s `LeadPicker` module with a preset distance threshold among the designs with the desired property. This metric combines the evaluation of diversity and accuracy; the higher, the better. The highest number of successful design clusters obtained across $\lambda$ values (task arithmetic), or training subsets (supervised finetuning) was reported for each property. In general, models trained on randomized SMILES yielded more successful design clusters than their canonical counterparts (Fig. A4) as also observed elsewhere (Arús-Pous et al., 2019). Due to the better performance of randomized SMILES, in what follows, we report only the results for randomized SMILES and include results for canonical SMILES strings in the Appendix.

Across design tasks, MTA yielded a higher number of successful design clusters than pretraining and finetuning (9.5K and 11.7K more on average, respectively; Fig. 2a and Fig. A5a ). Except for the '↑ No. Rings' task with randomized SMILES, the performance gain by MTA over supervised finetuning is statistically significant across datasets (Mann-Whitney U test, p-value < 0.01). This is a *striking finding:* despite using *no* positively labeled molecules, molecular task arithmetic can design a higher number of successful diverse molecules than models fine-tuned on up to 1024 positive compounds. This demonstrates the potential of task arithmetic for generative drug discovery without *any* positively labeled molecular examples.

To further shed light on the success of MTA, we inspect the success rate (ratio of designs satisfying the task; Fig. 2b and Fig. A5b) and number of total clusters (Fig. 2c and Fig. A5c). Results show that

MTA creates, on average, 24.4K more diverse molecule designs than the finetuned models and 7.4% more accurate designs than pretraining, while finetuning outperforms MTA by 14.1% in success rate. This is, on one hand, expected, since finetuning has explicit access to molecules fulfilling the desired property. On the other hand, it is surprising that MTA can create more clusters with less successful designs. Together, these findings display that MTA can preserve the diversity gained by pretraining during conditioning, while finetuning 'forgets'. Forgetting with finetuning is observed in different deep learning contexts (Vieira et al., 2024; Lampinen et al., 2025), and here we show that MTA is a technique that can maintain higher diversity while conditioning molecule designs.

**Distribution shifts.** To study how each training strategy shifts the property distribution, we first assess extrapolation capabilities for target properties, i.e., whether they can design molecules with property values outside the training sets. We inspected high-value design tasks with randomized SMILES and computed the maximum value of the task properties for the pretraining set and the designs (Table A2). MTA designs possessed values equal to the theoretical maximum or above the pretraining values in four of five cases (except for logP). In contrast, finetuning could saturate only one design task (Frac. $sp^3$C). This finding shows that molecular task arithmetic can extrapolate beyond the property distributions on which it was trained, unlike supervised finetuning, offering potential to explore new portions of the chemical space.

We next study how MTA and finetuning impact 'off-target' properties. We computed the distributional distance between off-target properties of the designs and the pretraining set. MTA designs possessed a smaller distance to the pretraining set than finetuning ones in general (Fig. A6, A7, A8), showing that MTA enables a more controlled steering of the models in the desirable portions of the chemical space (more details in the Appendix).

**Out-of-distribution generalization.** How does molecular task arithmetic perform when no positive molecule is available in the pretraining set? Here we seek an answer to this question via six design tasks across three descriptors. We picked fraction of $sp^3$C, logP, and TPSA, and pretrained three LSTMs (0.41M parameters) on 250K molecules from ChEMBL, such that (a) 0.2 $\leq$ Frac. $sp^3$C $\leq$ 0.85, (b) $1 \leq$ logP $\leq 6$, and (c) $50 \leq$ TPSA $\leq 100$, respectively. We defined design tasks as designing molecules whose properties are outside the pretraining set limits. We applied MTA on the molecules of the 'negative' task ($0.10 \leq \lambda \leq 1.00$; step size 0.10), i.e., training sets containing no molecules possessing the desired property. Fine-tuning with 'positive' molecules was also performed to assess performance when positive data are available. The number of successful design clusters was measured (with 100K designs) in increasing dataset sizes from 10 to 1024, across five repeats with different negative/positive molecule sets.

MTA designs 10K-50K successful molecule clusters across six design tasks (Fig. 3), although it is neither pretrained nor finetuned on *any* positive molecules. This represents a remarkable increase compared to the 1K-5K clusters obtained by the pretrained model, demonstrating the performance boost of MTA even in OOD settings. When 1024 positive molecules are available, finetuning achieves up to 55K successful design clusters. This highlights the added value of positive data in pushing the pretrained model toward a new region of the property distribution; MTA on negative data constitutes a valuable alternative. In fact, our analysis shows approximate "positive-to-negative data exchange rates", which depend on the target property. For 'Frac. $sp^3$C' and 'logP', even ten positive molecules are enough to outperform 1024 negative

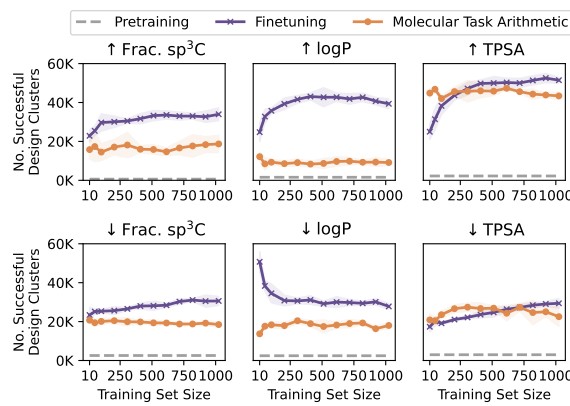

Fig. 3: *Out-of-distribution design*. The number of successful design clusters was computed at increasing sizes of 'negative' and 'positive' training sets. Mean (solid lines) and standard deviation (shaded areas) are reported (100,000 designs, five training-validation splits).

molecules, while for '↑ TPSA', at least 250 positive molecules are needed to match the performance of ten negative ones. For '↓ TPSA', 50 negative molecules can yield the same number of successful design clusters as 500 positive molecules, demonstrating that the value of negative data can even be higher than positive molecules in certain low-data settings. Taken together, our results show that MTA can design positive molecules via negative ones, even when no positive molecules exist in the pretraining set. Positive data remains, in general, valuable for out-of-distribution generalization (whenever available), while negative data can yield better performance for certain tasks in the low-data scenarios.

**Dual-objective design.** We next explore the capabilities of molecular task arithmetic for zero-shot dual-objective de novo design. This objective is central in drug discovery, for instance, for multi-target drug design, or selectivity optimization. To apply MTA in a dual-objective setting (Fig. 1b), we sum the scaled vectors yielding the best performance for each single task (Eq. 3) and scale the final vector in increasing $\lambda$ values (0.05 to 1.00 with a step size of 0.05). For finetuning, we finetuned the models on the training set of the single tasks sequentially (using positive molecules), as in recent literature (Ballarotto et al., 2023), with both task permutations. We reported the scores for the $\lambda$ and permutation yielding the highest number of successful design clusters for MTA and finetuning.

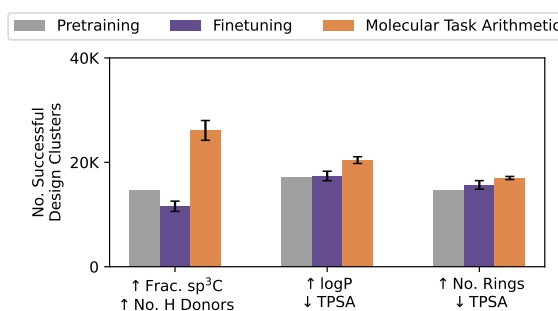

MTA obtained the highest average number of successful design clusters across all experiments (Fig. 4a, A9a), using no labeled positive data for either tasks. Mann-Whitney U test indicated statistical significance (p-value < 0.01) of the observed difference for all tasks except for '↑ *no. Rings* - ↓*TPSA*'. Inspecting the success rate and diversity (Fig. A10): sequential finetuning lowers the diversity of the designs by up to 20K clusters. While the designs of finetuned models achieve higher success rates in four out of six cases, the drop in diversity lowers the number of successful clusters. This hinders chemical space exploration and, thus, the potential to invent new chemistry with finetuning. MTA, in contrast, can better combine diversity and accuracy aspects, as in single-property tasks, and offers a promising approach for zero-shot multi-objective drug design.

Fig. 4: *Zero-shot dual objective molecule design.* Models are trained on randomized SMILES representation with sequential supervised finetuning and molecular task arithmetic to design molecules that possess two task properties simultaneously (average and standard deviation across five splits).

## 5.3 Effect of Task Vector Scaling and Practical Implications

Here, we further dig into molecular task arithmetic, to study the impact of $\lambda$ (Eq. 4) on the performance. The parameter $\lambda$ scales the task vector length and determines how far the task model 'lands' from the pretrained model after task negation. We compute validity and success rate across all 20 design tasks with 19 increasing $\lambda$ values (from $0.05$ to $1.00$ with a step of $0.05$). We measured the validity and success rate in those increasing lengths of task vectors, to reveal how much the model can move in the weight space before its generation capabilities collapse.

Across experiments, the validity decreases with increasing $\lambda$ values (Fig. 5 and Fig. A11). We explain this by the objective of the pretraining task: since maximizing next token prediction accuracy optimizes weights for generation validity, moving the model away causes a drop. The success rate also displays consistent patterns (Fig. 5 and Fig. A11): it first increases, then plateaus (typically for $0.30 \leq \lambda \leq 0.50$), and monotonically decreases afterward. This shows that molecular task arithmetic gradually conditions the pretrained model as the model moves along the task direction, and then, generative capabilities start limiting the number of successful designs. The consistent behaviors offer a practical insight to tune $\lambda$ for molecular task arithmetic models. Models can be edited by setting $\lambda = 0.30$ and gradually increasing the value until the success rate starts decreasing. Thanks to the peaking behavior, this heuristic can find the $\lambda$ value optimal for the task.

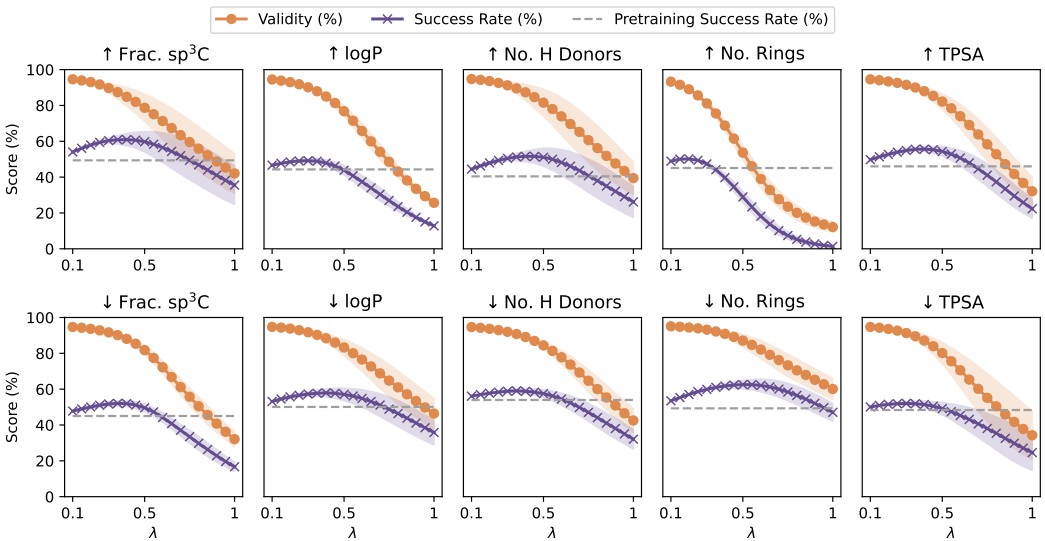

Fig. 5: *Task vector scaling*, across 19 increasing scaling factors ($\lambda$; Eq. 4). For each $\lambda$, validity and success rate for 100,000 designs were computed (average and standard deviation; five training splits).

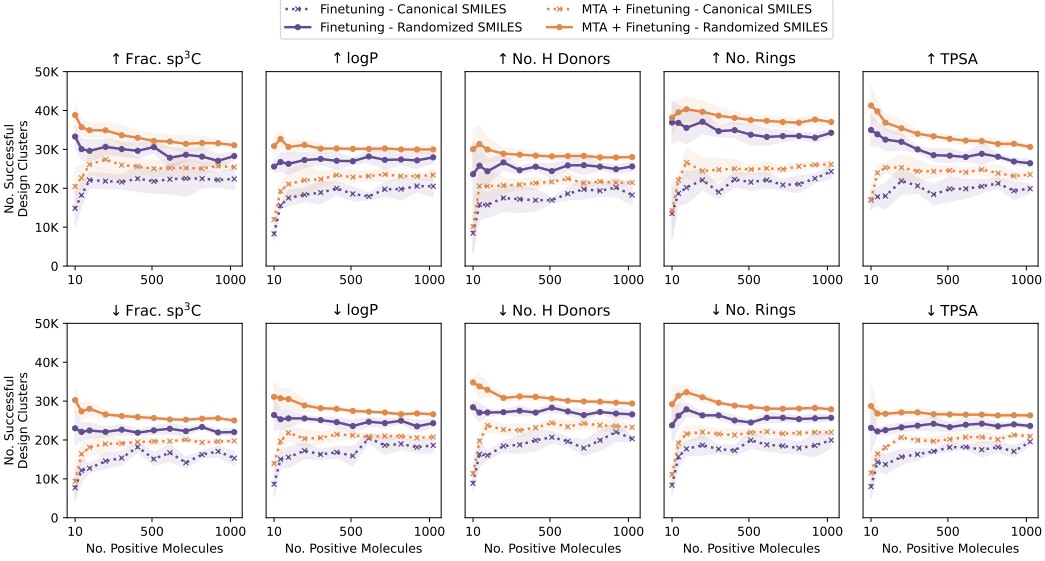

Fig. 6: *Few-shot molecule design.* Models are trained on an increasing number of positive molecules. Molecular task arithmetic (MTA) was first applied using negative data, and then the resulting model was finetuned with the available task data. Average and standard deviation across five splits reported.

## 5.4 BOOSTING FEW-SHOT MOLECULE DESIGN

We next delve into another interesting problem in drug discovery: few-shot molecule design. In few-shot molecule design, some positive labeled molecules are already available, which are used for molecule design. Additionally, negative molecules are usually available (and in general more abundant), but are not leveraged in traditional finetuning pipelines. Here we set out to investigate whether molecular task arithmetic (MTA) can allow leveraging negative molecules alongside positive ones to boost the capacity of models to generate molecules with desirable properties.

We applied MTA as in the zero-shot design experiments (Eq. 4) first, and then fine-tuned the obtained model with the known positive molecules (Fig. 1c). In other words, this pipeline replaces the

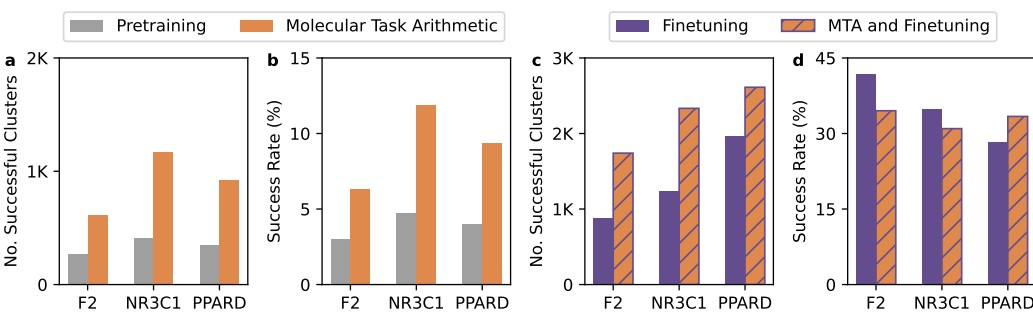

Fig. 7: *Bioactive molecule design*. Models are trained with well- and poor-docking molecules. 10,000 designs are generated with each strategy and 10,000 pretraining molecules are included as a control. Number of successful clusters (**a**, **c**) and success rate (**b**, **d**) are computed for each set.

pretraining model used for supervised finetuning with the MTA model, to harness the benefits of abundant and structurally diverse negative molecules. We compare the proposed approach to traditional supervised finetuning by computing the number of design clusters when increasing the number of positive molecules.

Across 20 tasks and the number of positive molecules studied, MTA surpassed the number of successful clusters obtained with finetuning, up to a 10K increase (Fig. 6). A deeper look reveals that MTA can design at least as many successful designs as supervised finetuning (Fig. A12), and yet maintain the design diversity (Fig. A13). Overall, these experiments point to a synergy: whenever positive molecules are available, MTA can be combined with supervised finetuning to increase the number of diverse hits. MTA can reach maximum performance even with fewer than 50 positive molecules. The number of clusters gradually decreases as the model is finetuned with further molecules. Such performance fluctuations, however, have a smaller magnitude than vanilla finetuning, underlining the robustness of MTA. Taken together, the experiments highlight MTA as a data-efficient, robust, and well-performing transfer learning strategy for few-shot molecule design. Combined with its simplicity, MTA can potentially become the new standard for transfer learning in de novo design, by allowing one to leverage both positive and negative data simultaneously.

## 5.5 BIOACTIVE MOLECULE DESIGN WITH INACTIVITY DATA

We now focus on designing molecules possessing a more complex property: a good docking score on a chosen protein target, starting from molecules that are *not* interacting with the target. Since ligand–protein interaction depends on shape complementarity, chemical compatibility, and binding-site dynamics, this setting presents a particularly challenging test case. We used Autodock-GPU (Eberhardt et al., 2021) for docking, and docked 100,000 randomly selected pretraining molecules against F2, NR3C1, and PPARD. We found that well-docking molecules constitute less than 5% for each target (Fig. A3), highlighting the scarcity of positive molecules and the difficulty of this task.

We evaluated molecular task arithmetic (MTA) to design well-docking molecules in zero- and few-shot settings. In the *zero-shot setting*, MTA yielded 2.3-2.8 times more successful molecule clusters compared to the pretraining set (10,000 control molecules) (Fig. 7a), with better docking scores (2.1-2.5 fold improvement, Fig. 7b). In the *few-shot setting*, MTA created 643 to 1100 more successful design clusters than traditional finetuning (Fig. 7c), and its designs were structurally more diverse from the training set (Fig. A14), suggesting that task arithmetic can create a more diverse set of promising molecular candidates. Overall, fine-tuning shows a higher success rate (7.29% and 3.84% higher for F2 and NR3C1, respectively), whereas for PPARD, MTA boosts the success rate by 5.18%. Notably, PPARD has the smallest number of positive molecules (12 molecules; 13% of F2 and 9% of NR3C1; Table A3). Taken together, these results demonstrate that MTA can design molecules with complex, desirable properties without any such labeled data points (Fig. A15). Furthermore, MTA can increase the diversity of successful designs in general, with particular promise in low data settings. These findings further demonstrate the potential of task arithmetic to mitigate data availability bottlenecks and open exciting new frontiers for de novo molecule design.

### 5.6 Molecular Task Arithmetic for Protein Design with Transformers

To extend our findings beyond LSTMs and small molecules, we applied molecular task arithmetic (MTA) to a large pretrained transformer language model for protein design. We experimented with ProtGPT2 (Ferruz et al., 2022), a 738M-parameter model pretrained on amino acid sequences, to design proteins with more structurally-ordered regions. A high degree of intrinsic disorder in proteins leads to multiple possible 3D conformations, posing a substantial challenge for designing proteins with a specific fold (Listov et al., 2024). We curated proteins lacking well-defined 3D structures from DisProt (Aspromonte et al., 2024) as highly disordered (negative) examples and applied MTA. We benchmarked our approach against ProtGPT2 finetuned on a highly structured subset of the pretraining set.

The fraction of amino acids predicted to be ordered by IUPred3 (Erdős et al., 2021) increases from 80% to 90% with MTA (Table 1), and the fraction of designs with at least one globular domain increases from 88% to 92%, compared to the pretrained model. Finetuning on highly structured proteins yields 97% and 100% on these metrics, respectively, at the cost of increased redundancy. Finetuning designs have a redundancy of 35% at 30% identity, meaning that 35% of the designed sequences are similar to at least one other finetuning design over 30%. On the contrary, MTA exhibits lower redundancy values (22%), suggesting that this approach can design more diverse proteins. DisProt (training set of MTA) is the most redundant library in the comparison; yet, MTA enables exploring high-diversity regions. Our experiments corroborate the versatility and capacity of MTA to achieve accuracy and diversity also with this additional modality, and model architecture and scale.

Table 1: Ordered amino acid ratio, globular protein ratio, and redundancy across datasets and design methods. Best (bold) and second-best (underline) approaches are highlighted.

| Method | Ordered AA Ratio | Globular Ratio | Redundancy (@30%) |
|---|---|---|---|
| Pretraining Set | 83% | 88% | 10% |
| Finetuning Set | 97% | 100% | 7% |
| DisProt | 28% | 0% | 44% |
| ProtGPT2 | 80% | 88% | **12%** |
| Finetuning | **97%** | **100%** | 35% |
| **MTA** | 90% | 92% | 22% |

### 5.7 On Failure Modes of Molecular Task Arithmetic

A key parameter in MTA is the magnitude of the task subtraction, controlled by $\lambda$ (Eq. 4). Excessive values lead to a deterioration of SMILES validity (Fig. 5) and a corresponding drop in success rates. This indicates that "jumping too far" from the pretrained model can induce model collapse. We observe a strong correlation between validity and success rate, suggesting that validity can serve as a practical proxy for $\lambda$ selection when success rate is difficult to measure. 'Selective task arithmetic' (Bowen et al., 2024) (i.e., updating certain weights only), or alternative model architectures and representations could be explored to mitigate the validity drop. Moreover, MTA could extrapolate beyond the training data, albeit depending on the design task and the frequency of positive molecules in the pretraining set. This means that the extrapolation capacities will have to be analyzed on a case-by-case basis – posing potential constraints for properties that are difficult to compute. Finally, while MTA showed promising results for dual-objective design, combining task vectors of conflicting tasks might introduce task collapse, constraining its applicability in multi-objective settings.

## 6 Conclusion

We proposed a transfer learning strategy to sidestep the low-data bottleneck in drug discovery: molecular task arithmetic (MTA). MTA leverages the diversity and abundance of negative molecules and can design molecules zero-shot. Experiments across architectures, molecular entities, and task difficulties show that MTA can design more diverse hits than models trained on positive data. While we studied molecular task arithmetic as an alternative (and complementary) approach to supervised finetuning, preliminary results highlight its integration with goal-directed optimization using reinforcement learning as an exciting research direction (*see* Appendix and Table A6). Thanks to its simplicity and performance, we expect MTA to attract further research and become a prominent transfer learning strategy for drug discovery, a field where negative data is abundant.

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

APPENDIX

SMILES STRINGS

SMILES (Weininger, 1988) is the most popular string representation for chemical language modeling (Fig. A1a). They annotate the bonds, atoms, and branches in a molecular graph, starting from any non-hydrogen atom. This characteristic allows creating multiple SMILES strings per molecule, randomized SMILES, enabling data augmentation (Bjerrum, 2017). Canonicalization algorithms were proposed to obtain a single SMILES string from any molecule, aiding in duplicate detection (Brown et al., 2019). Both SMILES formats are commonly used for molecule design (Grisoni, 2023), with randomized SMILES strings yielding higher chemical space exploration (Arús-Pous et al., 2019).

For modeling purposes, the SMILES string represents a molecule $m$ as $s_1, \ldots, s_n$, where $s_i$ is the $i^{th}$ token in the molecular string. The next token prediction language modeling objective is then defined as: $\max \sum_m^M \sum_i \log P(s_i|s_{<i}; \theta)$, where $\theta$ is typically a long-short term memory network (Segler et al., 2018) or a transformer (Bagal et al., 2021), and $M$ is a dataset of SMILES strings (Fig. 1b).

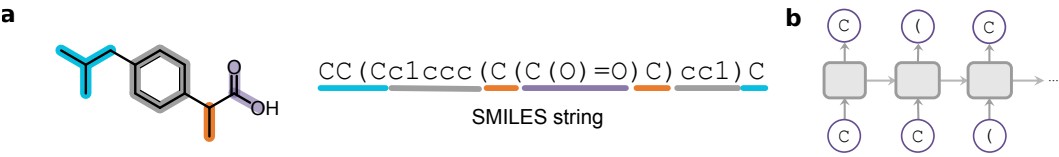

Fig. A1: *Molecular task arithmetic.* **(a)** SMILES strings annotate atoms by periodic table symbols and use additional tokens for bonds, branches, and rings. **(b)** Chemical language modeling casts SMILES generation as next-token prediction (e.g., using a sequence model, such as an LSTM).

EXPERIMENTAL SETTINGS

A chemical language model with an LSTM backbone is pretrained on a previously curated dataset of 1.5M drug-like molecules from (Gaulton et al., 2017; Özçelik & Grisoni, 2024). 100 models were trained for either canonical or randomized SMILES strings, with random hyperparameters (Table A4). For each representation, the model that generates the highest number of unique and new designs was selected for the finetuning stage.

The finetuning sets were curated from a previous study (Sun et al., 2017) by ensuring that no finetuning molecule is present in the pretraining set. The thresholds for high and low values are computed by rounding the median of the molecular property values among the pretraining molecules (Table A5 and Fig. A2). Five molecule sets are curated for each task (1024 train, 256 validation, 256 test molecules per split), where each molecule carries the task property. For models trained on randomized SMILES, 10-fold SMILES augmentation is used during finetuning. Early stopping on validation loss (cross-entropy) with a patience of five epochs and loss tolerance of $1 \times 10^{-5}$ is employed. 100,000 molecules were generated with each model.

Four metrics were computed to evaluate the quality of the de novo designs across the study:

- *Validity*: the percentage of 'chemically valid' molecules across designs. Chemical validity is checked by attempting to create a molecule object in `rdkit`, after filtering out empty string designs.

- *Success rate*: the frequency of distinct molecules outside the training sets that possess the task property. Molecules are deduplicated via canonicalizing the designs and keeping only one of the identical strings. Success rate evaluates the accuracy of the model (the higher, the better).

- *Number of clusters*: the number of clusters identified by the sphere exclusion algorithm. Tanimoto distance on extended connectivity fingerprints (Rogers & Hahn, 2010) is used with a threshold of 65% to identify the number of distinct clusters as suggested in previous

work (Xie et al., 2023). The metric is linked to #Circles (Xie et al., 2023), and the `rdkit` implementation is used. The higher the number of clusters, the more diverse the design library – a desired trait for drug discovery (Renz et al., 2024).

- *Number of successful molecule clusters*: the number of cluster centers that possess the task property. Successful designs are first extracted from the full list and then clustered to identify the structurally distant molecules. This metric combines the evaluation of diversity and accuracy, and is used as the main evaluation metric across the study. Higher values are preferred.

**Docking.** We selected coagulation factor II (F2), glucocorticoid receptor (NR3C1), and Peroxisome proliferator-activated receptor delta (PPARD) proteins for docking experiments since they are studied in drug discovery contexts and docking yields high enrichment for these targets (García-Ortegón et al., 2022). The binding site annotations, binding ligands, and non-binding ligands are curated from previous work (García-Ortegón et al., 2022). Any ligand that contains atoms besides C, H, O, N, S, P, F, Cl, Br, I, or is present in the pretraining set is filtered out. Ligands with 10-40 non-hydrogen atoms and a canonical SMILES length below 80 are extracted. Salts, charges, and stereochemistry annotations are removed. Meeko is used to preprocess the proteins and ligands for docking. Autogrid4 is used to create interaction maps of the protein binding sites, and Autodock-GPU is used to run the docking simulations (Santos-Martins et al., 2021).

20 docking simulations are run for each ligand, and the lowest binding energy across the runs is used as the docking score. 100,000 pretraining molecules are randomly sampled and also docked against each target using the same pipeline. The rounded values that define the lowest and highest 5% of the pretraining docking score distributions are used as thresholds to define 'well-docking' and 'poor-docking' molecules (Fig. A3). The thresholds for F2 are found to be -10.75 and -5.75 for well-docked and poorly docked molecules, respectively. For NR3C1 and PPARD; the same thresholds are computed as -12.00 and -6.00.

The actives with a docking score below the well-docking threshold, and inactives with a docking score above the poor-docking threshold are extracted for supervised finetuning and molecular task arithmetic. The molecules in each set are represented with randomized SMILES and augmented 10 times. The models are trained with early stopping with configurations the same as above. 10,000 molecules are generated with each trained model, and novel and unique designs are docked against the corresponding target using the same docking pipeline used for datasets.

We train models with molecular task arithmetic ($0.20 < \lambda < 0.90$) using poor-docking molecules for zero-shot design. For the few-shot setting, we apply supervised finetuning using positive molecules and task arithmetic as previously (Section 5.4). We produce 10,000 samples with each strategy and dock the novel and unique designs. We randomly sample 10,000 molecules from the pretraining set as a control and compute the number of successful clusters and success rate.

### SIDE EFFECTS

A desirable trait of model conditioning is the possibility of having minimal side effects on the 'off-target' properties. Being able to minimize side effects would allow for a more precise steering of the model in the chemical space, by preserving desirable molecular properties (i.e., synthesizability, lack of toxicity, etc.). We quantify the side effects of both finetuning and molecular task arithmetic for all design tasks, on the remaining four descriptors. These descriptors have a low correlation with the 'target' properties (Table A1). For each task, we computed the Kolmogorov-Smirnov (KS) distance between the off-target molecular properties of the designs and the pretraining set (the lower the KS, the more similar the distributions). The usefulness of this analysis is corroborated by observing the KS distances of the designs obtained by pretraining (Fig. A7), which are all below 5%.

Across the ten design tasks for randomized SMILES, molecular task arithmetic showed a smaller mean distributional distance to the pretraining set in 32 of the 40 comparisons (Fig. A6), of which 15 were confirmed by a statistical test (Mann-Whitney U test, p-value $< 0.01$). On canonical SMILES strings, all models have a higher average distance (Fig. A8), possibly because canonicalization restricts the models to explore 'side-effect-free' portions of the chemical space. Overall, the analysis shows that molecular task arithmetic can combine its accuracy and diversity on the design task, with

less side-effect than the traditional finetuning approach. This suggests that task arithmetic allows for a more controlled steering of the models in the desirable portions of the chemical space.

### MOLECULAR TASK ARITHMETIC AND REINFORCEMENT LEARNING

We have run preliminary experiments to test the potential of molecular task arithmetic within goal-directed optimization with reinforcement learning. We have run five REINVENT loops with default parameters to design molecules with high and low TPSA values using pretrained and MTA models from the out-of-distribution design experiments (Section 5.2). With 100,000 molecules designed per run (Table A6), REINVENT with MTA obtained 15K and 35K more successful design clusters for low-tpsa and high-tpsa design tasks, respectively, than running REINVENT with the pretrained model. MTA also increased the success rate by 27% and 47%, while pretrained model yield 4K and 1K more design clusters in the respective tasks. While a systematic, large-scale analysis across optimization algorithms and design tasks is needed to accurately assess the harmony between MTA and RL, this preliminary analysis shows that MTA can also be used to kick-start an RL loop for zero-shot design with external reward functions.

Table A1: The correlations of molecular properties. Pearson correlations are computed over the pretraining set. Correlations for the tasks in dual-objective design experiments are marked with boldface.

| Property | No. H. Donors | No. Rings | logP | TPSA | Frac. sp$^3$ C |
|---|---|---|---|---|---|
| No. H. Donors | - | -0.09 | -0.28 | 0.17 | **0.02** |
| No. Rings | -0.09 | - | 0.40 | **0.26** | 0.13 |
| logP | -0.28 | 0.40 | - | **-0.12** | 0.08 |
| TPSA | 0.17 | **0.26** | **-0.12** | - | 0.57 |
| Frac. sp$^3$ C | **0.02** | 0.13 | 0.08 | 0.57 | - |

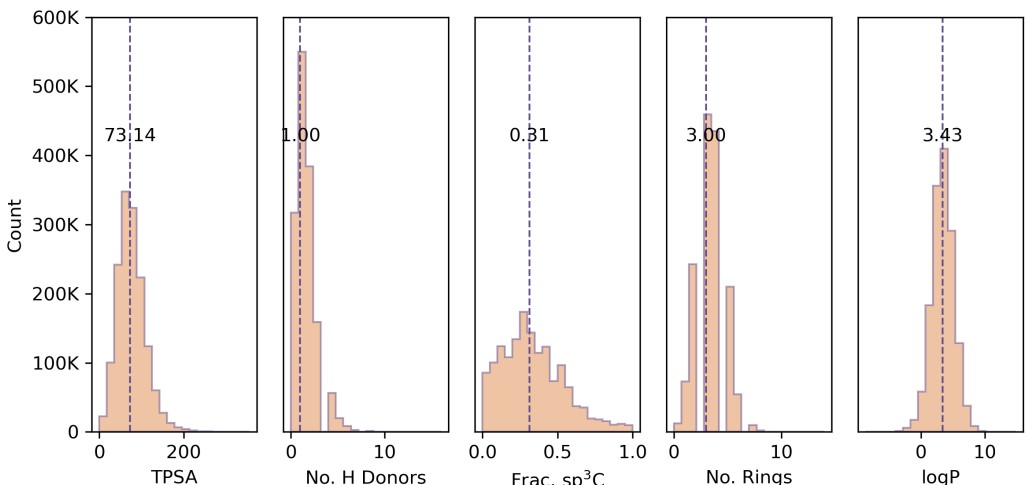

Fig. A2: Distribution of molecular properties in the pretraining set. The dashed line corresponds to the median, which is also annotated.

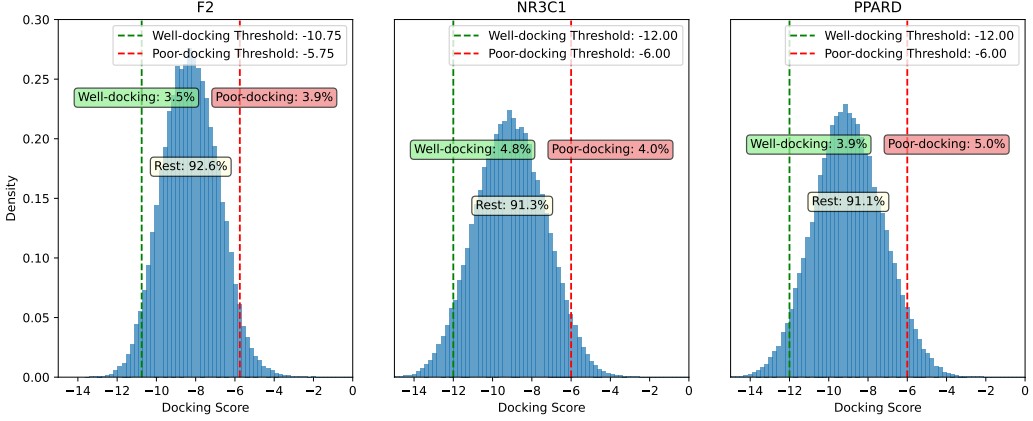

Fig. A3: Docking score distribution of 100,000 pretraining molecules for the studied proteins.

Table A2: Maximum values of molecular descriptors in the pretraining set and designs of supervised finetuning and molecular task arithmetic for each setup. Highest values are displayed in boldface.

| Task | Setup Index | Pretraining | Finetuning | Molecular Task Arithmetic |
|---|---|---|---|---|
| Frac. sp$^3$C | 0 | 1.00 | 1.00 | 1.00 |
| | 1 | | 1.00 | 1.00 |
| | 2 | | 1.00 | 1.00 |
| | 3 | | 1.00 | 1.00 |
| | 4 | | 1.00 | 1.00 |
| logP | 0 | 14.79 | 11.45 | **17.58** |
| | 1 | | 9.25 | **14.85** |
| | 2 | | 11.86 | **14.80** |
| | 3 | | 12.87 | **16.32** |
| | 4 | | 9.69 | **15.34** |
| No. H Donors | 0 | **16** | 9 | 15 |
| | 1 | | 9 | 15 |
| | 2 | | 11 | 14 |
| | 3 | | 10 | 15 |
| | 4 | | 9 | 15 |
| No. Rings | 0 | 14 | 10 | **16** |
| | 1 | | 11 | **17** |
| | 2 | | 12 | **20** |
| | 3 | | 11 | **14** |
| | 4 | | 10 | **17** |
| TPSA | 0 | 356.28 | 217.24 | **431.13** |
| | 1 | | 252.19 | **431.79** |
| | 2 | | 230.99 | **408.87** |
| | 3 | | 240.80 | **413.41** |
| | 4 | | 258.06 | **433.85** |

| Protein | Positive | | Negative | |
| | Training | Validation | Training | Validation |
|---|---|---|---|---|
| F2 | 140 | 47 | 3208 | 1070 |
| NR3C1 | 90 | 31 | 525 | 176 |
| PPARD | 12 | 5 | 570 | 190 |

Table A3: Dataset sizes used for bioactive molecule design experiments.

Table A4: *Hyperparameter space for pretraining the LSTMs.* 100 models are trained by randomly subsampling the hyperparameter space. NVIDIA A100 GPUs with 40GB of memory are used. Training one model on 1.5M SMILES strings took approximately 12 hours on average. Models with different parameters are trained simultaneously by using one GPU per model in a supercomputer.

| Hyperparameter name | Space |
|---|---|
| Number of layers | 1, 2, 4, 6, 8 |
| Hidden state dimension | 256, 512, 1024, 2048 |
| Dropout rate | 0.0, 0.1, 0.15, 0.2, 0.25 |
| Vocabulary size | 33 |
| Input sequence length | 82 |
| Learning rate | $1 \times 10^{-4}, 5 \times 10^{-4}, 1 \times 10^{-3}, 5 \times 10^{-3}, 1 \times 10^{-2}$ |
| Maximum number of epochs | 1000 |
| Batch size | 8192 |
| Optimizer | Adam |

Table A5: *Thresholds used to define the design tasks.* Values less than or equal to the threshold are considered low, and molecules with higher values are labeled high.

| Property | Threshold |
|---|---|
| Fraction of sp$^3$-hybridized carbons | 0.3 |
| Number of hydrogen bond donors | 1 |
| Number of rings | 3 |
| Octanol-water partition coefficient (logP) | 3.5 |
| Topological polar surface area (TPSA) | 75 |

Table A6: Number of successful design clusters, success rate, and number of clusters for reinforcement learning experiments with five different seeds. Average and standard deviation of each metric across runs are reported with 100,000 designs each.

| Task | Method | No. Successful Design Clusters | Success Rate | No. Clusters |
|---|---|---|---|---|
| Low TPSA | Pretraining | $7542 \pm 696$ | $12\% \pm 1\%$ | $\mathbf{36245 \pm 1867}$ |
| | **MTA** | $\mathbf{22459 \pm 912}$ | $\mathbf{39\% \pm 2\%}$ | $32419 \pm 1499$ |
| High TPSA | Pretraining | $1339 \pm 137$ | $1\% \pm 0\%$ | $\mathbf{43620 \pm 1197}$ |
| | **MTA** | $\mathbf{36492 \pm 851}$ | $\mathbf{48\% \pm 1\%}$ | $42304 \pm 586$ |

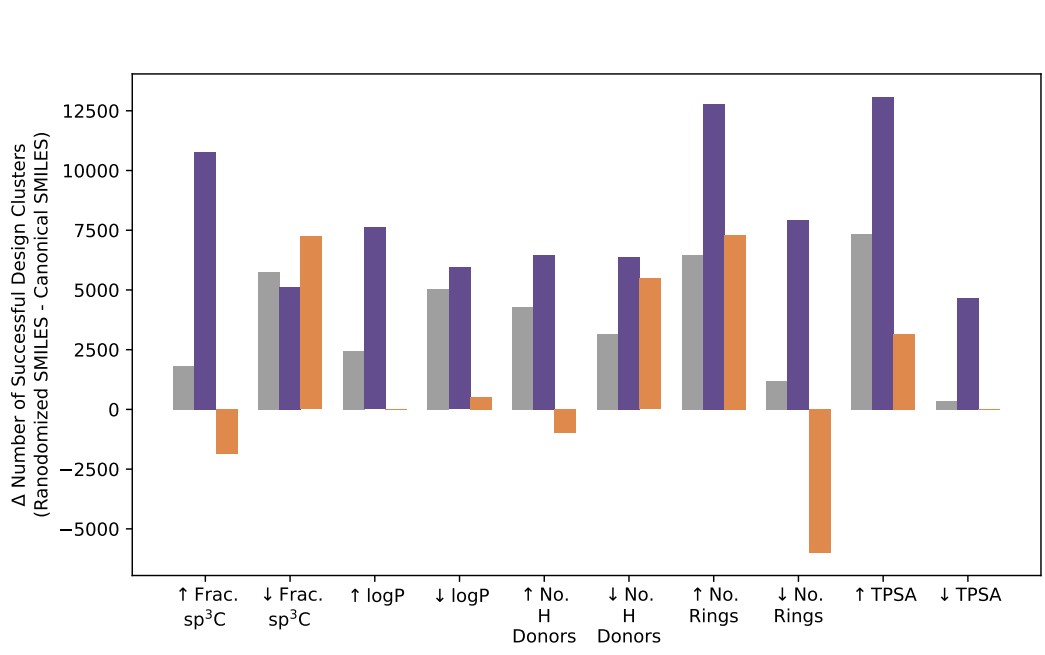

Fig. A4: *Comparison of SMILES representations.* The average number of successful design clusters is computed per design task using canonical and randomized SMILES representations. Bar heights report the scores obtained by using canonical SMILES subtracted from using randomized SMILES.

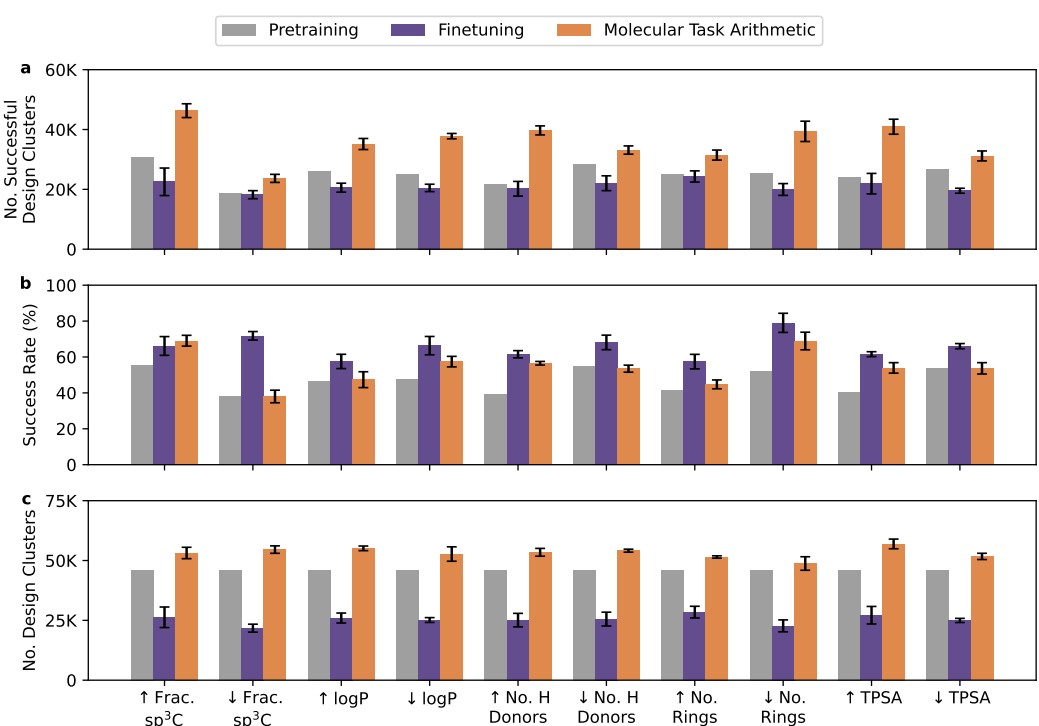

Fig. A5: *Zero-shot molecule design with molecular task arithmetic.* Models were trained on 10 design tasks of canonical SMILES strings, with molecular task arithmetic and supervised finetuning. 100,000 molecules were designed and clustered. **(a)** The number of cluster centers that possess the desired properties, **(b)** ratio of the designs that satisfies the design task, and **(c)** number of clusters were computed. The pretrained model is also included in the analysis as a baseline. Bar heights report the mean statistics across five training sets, and error bars denote the standard deviation. Finetuning was conducted up to the training sets of 1024 molecules, while molecular task arithmetic used 10,240 negative molecules and no labeled positive data. Yet, molecular task arithmetic obtained higher number of successful design clusters across design tasks.

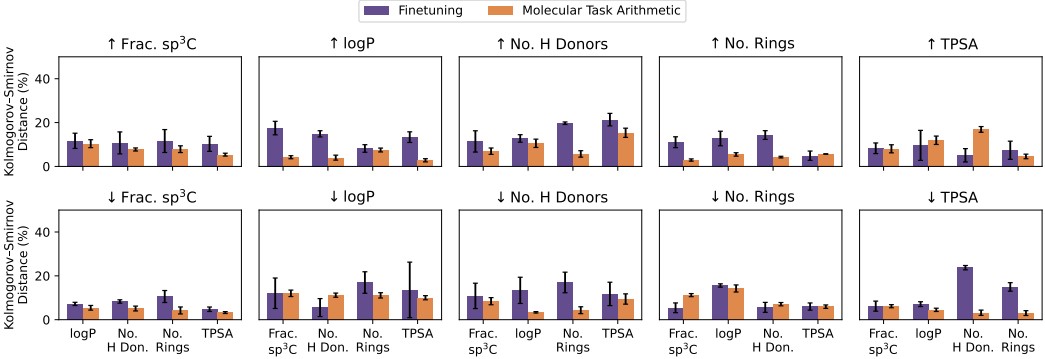

Fig. A6: *Side effects of transfer learning strategies.* Kolmogorov-Smirnov distance between the molecular properties of the designs on SMILES tasks and the pretraining set is computed for off-target properties. Lower distance to the pretraining set indicates that the conditioning caused less change in non-conditioned molecular properties. Bar heights display the means across five training splits, and error bars display the standard deviations.

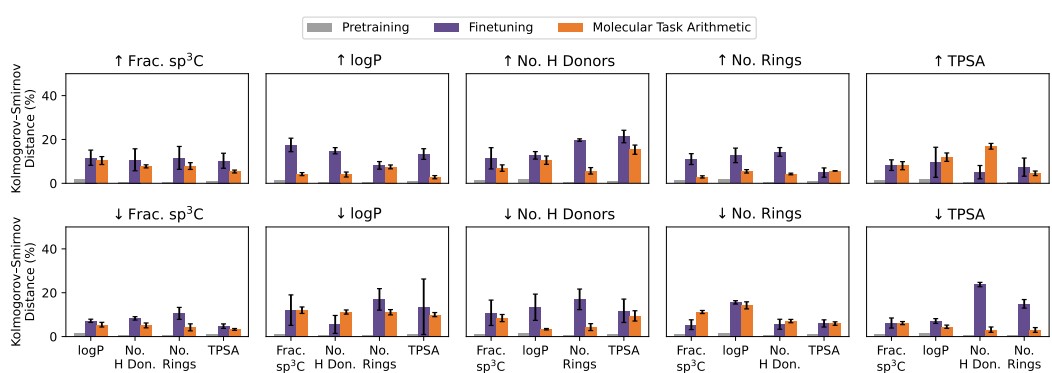

Fig. A7: *Side effects of conditioning strategies and pretraining.* Compuations are described in (Fig. A6). Bar heights display the means across five training splits, and error bars display the standard deviations. In 32 of 40 comparisons, molecular task arithmetic caused less side-effects.

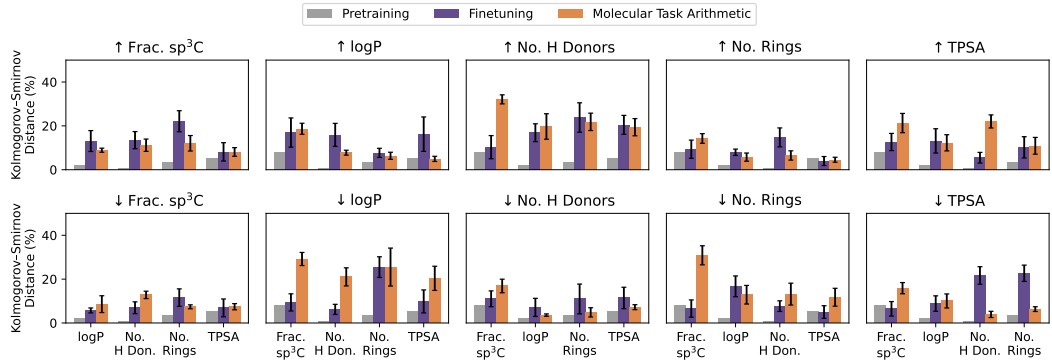

Fig. A8: *Side effects of transfer learning strategies on canonical SMILES experiments.* Kolmogorov-Smirnov distance between the molecular properties of the designs on canonical SMILES tasks and the pretraining set is computed, except for the task property. Bar heights display the means across five training splits, and error bars display the standard deviations.

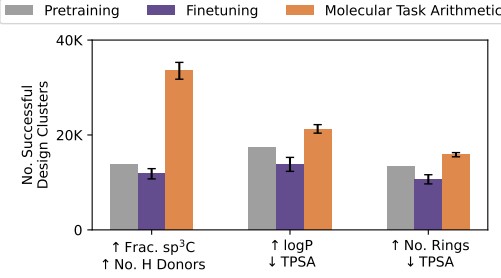

Fig. A9: *Zero-shot dual objective molecule design on canonical SMILES.* Models are trained on canonical SMILES representation with sequential supervised finetuning and molecular task arithmetic to design molecules that possess two task properties simultaneously. Number of successful design clusters, success rates, number of clusters, and validity are measured. Experiments are repeated five times, and the mean (bar height) and standard deviation (error bar) are reported.

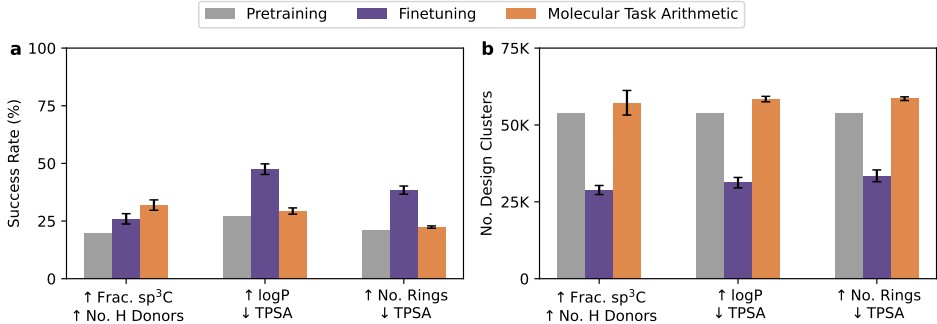

Fig. A10: *Zero-shot dual objective molecule design on canonical SMILES.* Models are trained on randomized SMILES representation with sequential supervised finetuning and molecular task arithmetic to design molecules that possess two task properties simultaneously. Number of clusters and success rates are measured. Experiments are repeated five times, and the mean (bar height) and standard deviation (error bar) are reported.

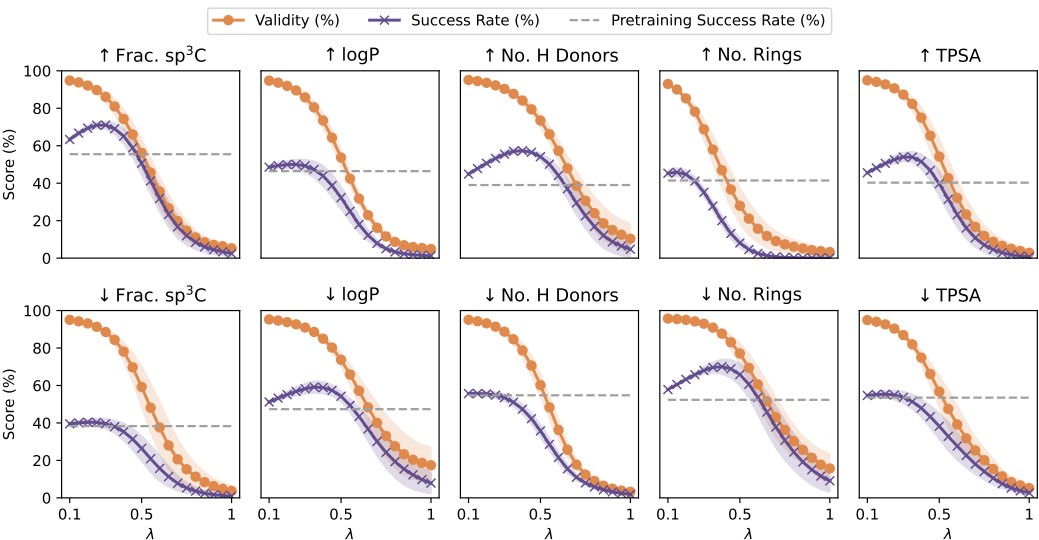

Fig. A11: *Impact of molecular task vector scaling on canonical SMILES.* For each design task defined on canonical SMILES strings, molecular task arithmetic is applied in increasing scaling factors ($\lambda$; Eq. 2). 100,000 molecules are designed with all $\lambda$s and validity and success rate are measured across five training splits. Lines denote the means across runs, and shaded areas describe the standard deviations.

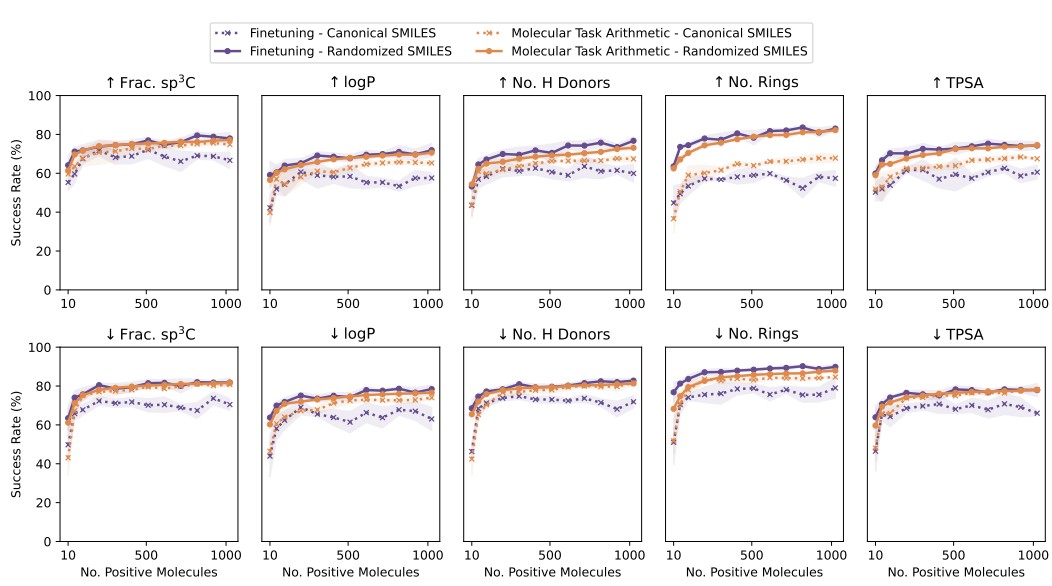

Fig. A12: *Success rates for few-shot molecule design.* Experimental pipeline is detailed in (Fig. 6). The mean and the standard deviation of the number of successful design clusters across five splits are computed (lines and shaded regions, respectively).

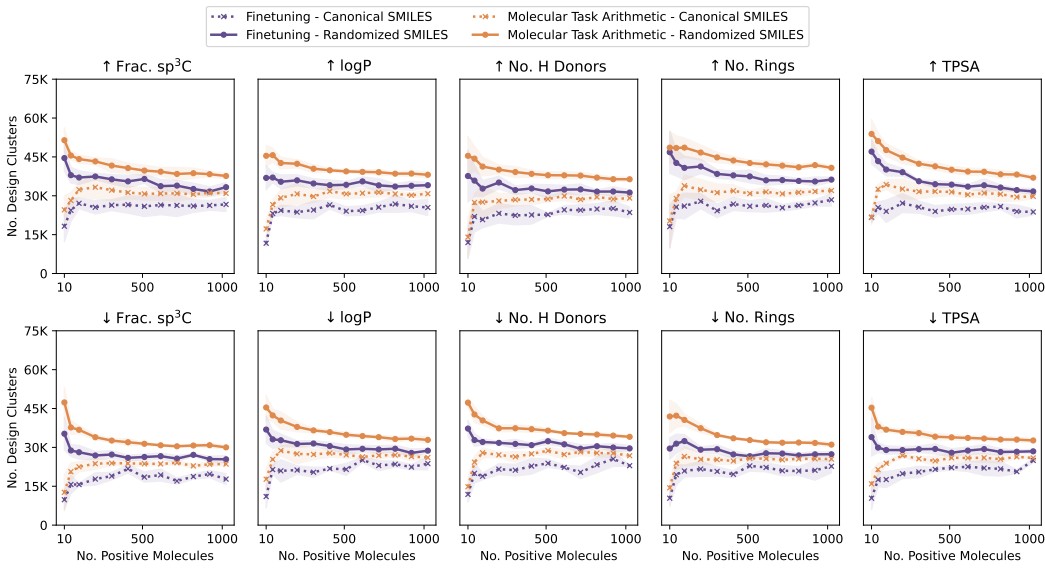

Fig. A13: *Number of design clusters for few-shot molecule design.* Experimental pipeline is detailed in (Fig. 6). The mean and the standard deviation of the number of successful design clusters across five splits are computed (lines and shaded regions, respectively).

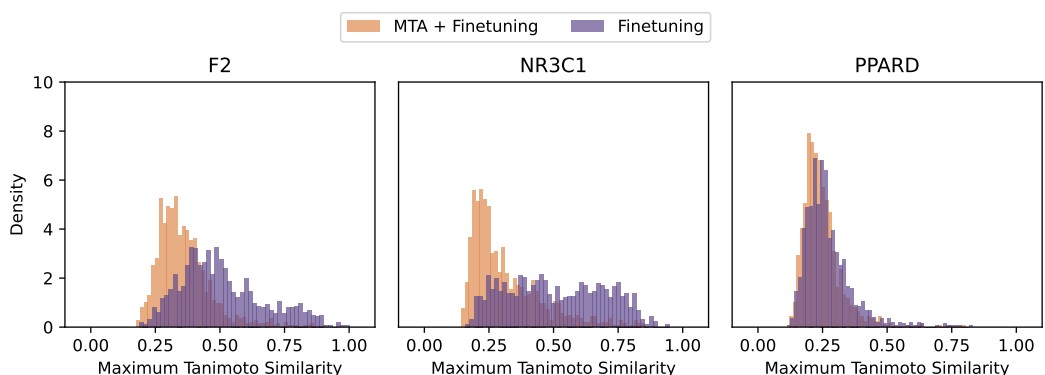

Fig. A14: Structural similarity distributions of top-1000 scoring molecules for finetuning and molecular task arithmetic in few-shot settings. Structural similarity is computed as Tanimoto similarity of extended connectivity fingerprints (radius=2, nBits=2048).

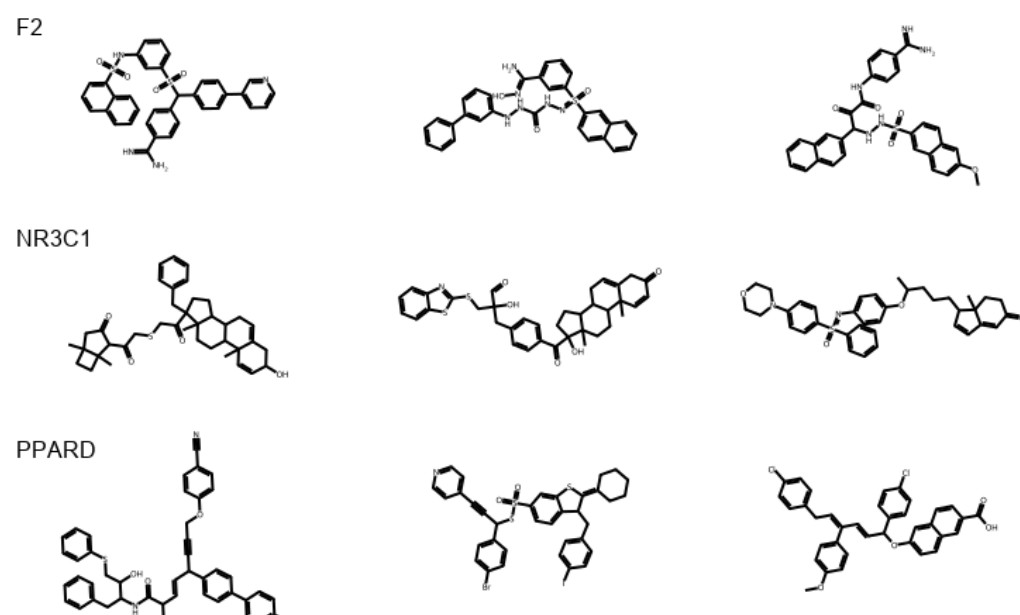

Fig. A15: Top-3 molecules designed by molecular task arithmetic in few-shot bioactive molecule design experiments, according to $\Delta G$. Each row corresponds to designs for a different protein target.

