# OpenReview forum: "Look the Other Way: Designing 'Positive' Molecules with Negative Data via Task Arithmetic"
_ICLR.cc/2026/Conference — Submitted to ICLR 2026_

### Official Review · Reviewer_ejE6 · 2025-10-31

**Soundness:** 2
**Presentation:** 3
**Contribution:** 3
**Rating:** 2
**Confidence:** 4

**Summary:**

This paper presents an exploration of the application of task arithmetic in molecular generation tasks. Results shows that via task arithmetic, the success rate of generating positive molecules can be improved over merely pre-training and fine-tuning over positive molecules.

**Strengths:**

- This paper is an important initial exploration on using task arithmetic for positive molecule generation. The task setting is meaningful and may be a future way for molecule designing or optimization.
- The proposed method can beat pre-training and finetuning in generaing more clusters of successful molecules.

**Weaknesses:**

- The task setting is a bit over-simplified. The experimental setting is no positively LABELED molecules, instead of no positively molecules at all. Actually, a large amount of positive molecules exists in the pre-training data, according to Figs A1 and A2. This is also why in Fig. 2, even a merely pre-trained model (grey color) presents a good success rate. I would expect a setting where positive molecules never appeared in both pre-training and fine-tuning, i.e., the results of 'pre-training' should be 0 or close to 0, then the proposed method to be explored in this setting, so people can know if task arithmetic really works for 0-shot design or not.
- This paper is largely not self-contained. Many tables or figures mentioned in the main text turned out to be in the appendix, which harms the readability.
- The definition of successful "clusters" is unclear and not mentioned.

**Questions:**

- How do you define/find "successful clusters"?

---

> ### Author Response · Authors · 2025-11-24
> **Response I to Reviewer ejE6**
>
> We thank the reviewer for highlighting the potential of our work for molecule design.  In the revised version of the manuscript, we addressed all doubts of the reviewer with experiments, and our new analysis confirms the potential of MTA.
>
> 1. **Zero-shot design.** We thank the reviewer for this suggestion and added a new section on out-of-distribution generation with task arithmetic, in other words, when no positive molecules are present in either the pretraining or the finetuning set. Our results show that task arithmetic can improve the number of successful clusters compared to pretraining only, up to  45K in some tasks. The results of this analysis are available now in Section 5.2.
>
> 2. **Content.** Due to space limitations imposed by ICLR, we had to move many of our analyses that were secondary to the Appendix. We understand that this could impact readability. If the reviwer has a particular figure that they see fit for the main text, we would be happy to consider re-arranging the content.
>
> 3. **Cluster definition.** We have clarified the meaning of “clusters” in the main text, that is, the number of clusters the LeadPicker algorithm identifies given a distance threshold
>
> ### Answers to Questions
>
> 1. **Cluster definition** This is now defined in the main text as follows: the number of clusters among the designs with the desired property (as identified by the LeadPicker module in rdkit
>
> We hope these to answer all the questions and concerns of the reviewer on our work.

---

### Official Review · Reviewer_nmQg · 2025-10-31

**Soundness:** 3
**Presentation:** 3
**Contribution:** 3
**Rating:** 6
**Confidence:** 4

**Summary:**

To train models to generate molecules with desirable properties, high-quality examples of molecules with such properties are often required. But such "positive" examples may be nearly non-existent in the general training population and extremely difficult to curate.  The authors propose molecular task arithmetic, a transfer learning technique to fine-tune neural networks with only "negative" samples. Rather than fine-tuning on positive examples, the technique uses abundant "negative" data to fine-tune an opposing model. Moving the pre-trained models' weights in the opposite direction of this opposing fine-tuned model, the authors hypothesize, will cause it to generate molecules with the desired properties.

**Strengths:**

1. The idea of going in the opposite direction from the "bad" model is very interesting and innovative. Being able to unlock potentially new behavior from entirely unseen domains could prove to be of huge significance and could open new avenues of research and application. The proposed solution is very elegant.
2. The central premise of the lack of "positive" samples in scientific domains, especially drug design, is a significant problem.
3. It is very interesting that the model obtains good scores on challenging tasks like improving docking score as the negative examples are not necessarily well defined in such a case.

**Weaknesses:**

1. "Current training strategies rely on rare, positive molecules." This is not true in general. This may be true for the case of fine-tuning approaches. But there is a long line of work in latent space and post-hoc optimization (such as Bayesian optimization, generative algorithms, or gradient-based steering) for generative design. So will this may be a strong candidate for a transfer learning strategy for molecule design models; the aforementioned techniques that do not require transfer learning or new data may be a more practical approach.
2. According to figure 4, it seems like there is a significant decrease in validity as the $\lambda$ is increased, as a result, finding the optimal model with sufficiently high validity and success rate may be challenging to find, and casts doubt on how steerable these models may actually be.

**Questions:**

1. How is a randomized SMILES different from a regular SMILES representation of a molecule? Is it a result of data augmentation?
2. Are there differences in performance depending on the model architecture or size of the model being used? As this is a weight update technique, I would be very curious as to how different models behave with this approach, if any trends are interesting
3. What is the difference between the number of successful design clusters and the number of design clusters in Figure 2?

---

> ### Author Response · Authors · 2025-11-24
> **Response I to Reviewer nmQg**
>
> We thank the reviewer for calling our work "elegant", potentially "of huge significance", and emphasizing the magnitude of the data scarcity problem this work is targeting. We address their two concerns below and answer their questions.
>
> 1. **Scope clarification.** We thank the reviewer for the clarification. We indeed propose task arithmetic as an approach to learn from negative data and study its performance to supplement or substitute finetuning for conditional distribution learning. The cited approaches all require reward/fitness functions, which may or may not use positive/negative data, and are more suitable for goal-directed optimization tasks. The revised manuscript clarifies the scope with the following edits:
>
>    - Current *training* strategies rely on these rare, 'positive' molecules -> Current *finetuning* strategies. (Section 3)
>    - outperforming traditional *transfer learning* in many design tasks -> ouperforming traditional *finetuning* (Conclusion)
>    - Added the following: While here we studied molecular task arithmetic as an alternative and supplement to supervised finetuning, its integration with goal-directed optimization with genetic algorithms and reinforcement learning is an exciting research direction. (Conclusion)
> 2. **Validity analysis.** In our large-scale analysis, we did not find examples where this would be the case, i.e., there was always a nice trade-off point between conditioning and validity. Nonetheless, for the sake of completeness, we have added a new section on possible failure modes (Section 5.7) where we reflect on potential issues related to the effect of lambda and propose "selective task arithmetic" as a future research direction to edit models with minimal side effects on validity.
>
> Thank you for these suggestions.
>
> ### Answers to Questions
>
> 1. **SMILES Formats.** Multiple SMILES strings can be obtained from a molecular graph, each starting with a different random node, or using a different traversal route. Canonical SMILES strings use fixed rules for initial node selection and the traversal. Randomized SMILES can indeed be used for data augmentation, and here we adopt a 10-fold augmentation approach during finetuning. The difference between the two representations was already described in the background section, and now we have added a citation that compares both representations in generative settings.
> 2. **Model scaling.** Thanks for introducing the model scaling aspect. Our initial models contained 3.17M parameters, and the revised manuscript contains experiments with three more LSTMs of 0.4M parameters and a pretrained transformer (ProtGPT2) with 738M parameters. While we do not analyze model scaling trends systematically – due to time and space constraints – we do see that the results we obtained with initially are applicable also to smaller and larger models. We briefly mention this behavior in Section 5.6.
> 3. **Cluster definition.** The number of design clusters is a proxy for the diversity of all designs, while the number of successful designs is computed only among the 'successful' designs, i.e., the designs that possess the task property. In the revised version of the manuscript, we clarify that LeadPicker module from rdkit is used to identify number of clusters. We thank the reviewer for bringing this point to our attention.
>
> With these responses, we hope to have addressed all of the concerns and questions by the reviewer, and to have better clarified the scope of our research.

---

### Official Review · Reviewer_ya8s · 2025-11-01

**Soundness:** 4
**Presentation:** 4
**Contribution:** 4
**Rating:** 8
**Confidence:** 4

**Summary:**

The paper applies the concept of task arithmetic to molecule generation. Task arithmetic describes the process of linearly combining the weights of a pretrained foundation model with those of a finetuned model. This works surprisingly well in the domain of molecule generation and property optimization, where “positive”/desirable examples to fine tune on are exceptionally rare, but “negative” examples are abundant. The method employed in the paper involves finetuning on the negative examples and then taking the difference of the model weights, to lead the foundation model away from the space of negative molecules.

**Strengths:**

Generally, the writing is very clear and communicates the arguments and ideas without issue, even to a less experienced reader. The concept of task arithmetic is very elegant, and its application to molecular optimization feels almost intuitive because of the quality of the explanations. The experiment design and results are very extensive, which additionally strengthens the main point of the paper. The graphs/tables are clear and easy to read.

**Weaknesses:**

The lack of mention of chemical LMs, like ChemBERTa or ether0, takes away from the credibility of the article. Though the results are impressive and interesting, LSTMs lack much of the expressive power of LMs. Though task arithmetic appears to work in the case of LLMs on natural language, it’s not clear whether that will extrapolate to models working on the chemical domain, especially given the apparent challenge with this generation task. Additionally,

**Questions:**

In section 5.2, what is a “cluster”? It does not appear to be defined anywhere previously in the article.
Line 282: “task arithmetic can be a technique to maintain [...]”. Maybe replace it with “task arithmetic is a technique that can maintain [...]” or even “task arithmetic is a technique that maintains [...]”.
It’s interesting that randomized SMILES inputs consistently perform better than canonical SMILES. Though not critical, discussion on this could be informative, if not out of scope.
Line 472: “[...] that leverages the diverse and abundant negative molecules”. Maybe replace it with “[...] that leverages the diversity and abundance of negative molecules”.

---

> ### Author Response · Authors · 2025-11-24
> **Response I to Reviewer ya8s**
>
> We thank the reviewer for finding our approach elegant; communication clear; and our experiments extensive. Below we address their only concern with further experiments.
>
> 1. **Testing pretrained models and other molecular modalities** Thanks for this great suggestion. The revised manuscript contains a new molecular design task (Section 5.6) with a large transformer model pretrained on proteins (ProtGPT2). Our results confirm the broad applicability of MTA to different models and molecular entities.
>
> ### Answers to Questions
>
> 1. We have clustered the molecules using rdkit's LeadPicker module, which uses the sphere exclusion algorithm to find the number of clusters with a given distance threshold. The implementation details are available in the appendix, and we clarified what a cluster is in the first occurrence (Section 5.2; paragraph 1).
> 2. Rephrased line 282 accordingly.
> 3. Added a reference to literature comparing randomized and canonical SMILES strings for generative modeling (doi.org/10.1186/s13321-019-0393-0)
> 4. Rephrases line 472 as suggested
>
> We hope these responses answer all the questions of the reviewer.

---

### Official Review · Reviewer_mQhA · 2025-11-01

**Soundness:** 1
**Presentation:** 2
**Contribution:** 1
**Rating:** 2
**Confidence:** 4

**Summary:**

This paper proposes a “look the other way” / molecular task arithmetic approach for de novo molecular design: instead of fine-tuning on scarce positive examples (molecules with desired properties), the model fine-tunes on abundant negative examples (molecules that fail the property), obtains a “negative task vector,” and then moves in the opposite direction in parameter space to generate candidate molecules that are more likely to satisfy the target property. The method is evaluated on a set of relatively simple property targets and a few docking targets, showing that (i) negative-only training can still produce valid, property-consistent molecules and (ii) in some settings it improves diversity compared to standard positive fine-tuning.

**Strengths:**

- Clever use of negative data. The key insight—most real medicinal chemistry data is overwhelmingly negative, so we should learn from what “doesn’t work” and then invert the direction—is both practical and underexplored. It directly speaks to the data imbalance that plagues many molecular design tasks.

- Empirically effective on some tasks. On the tested single-property and docking-based tasks, the method can recover a nontrivial number of “successful” and diverse solutions, sometimes outperforming standard fine-tuning in terms of successful clusters. This suggests the parameter-space editing idea is not just a theoretical curiosity but can actually steer generation.

**Weaknesses:**

- Property space is too easy / narrow. Most experiments are on relatively simple, monotonic, and well-behaved properties (e.g., logP high/low, TPSA high/low, counts of functional features). Even the docking objectives look like “flat-bottom” tasks—once you’re in a good-enough region, the signal stops being very discriminative. This makes it unclear whether the proposed method would still work on sharper, more structured, or conflicting objectives. The paper should include harder benchmarks such as those in GuacaMol (distribution-learning tasks, scaffold-aware objectives, isomer/chemotype control) or other community-accepted suites to demonstrate generality and push beyond easy scalar filters.

- No discussion of sample efficiency for molecular design. A main motivation of the approach is precisely that positive samples are scarce, yet the paper does not really quantify sample efficiency:

- Comparative baselines are underspecified. If the claim is “we can get good molecules without positives,” then the right comparison is not only vs. a naïve positive fine-tuning, but also vs. (i) property-conditioned generation trained on mixed labels, (ii) reinforcement-learning style optimization (REINVENT, PPO), and (iii) strong open benchmarks like GuacaMol’s goal-directed baselines. Right now, it’s unclear whether the gains come from the negative-task idea itself or from the fact that the tasks are relatively easy.

- Limited analysis of failure modes. Task arithmetic on model weights is a fairly aggressive operation; the paper should analyze when it fails: Does it hurt validity? Does it bias toward weird chemotypes? Does it collapse to trivial solutions when the negative set is heterogeneous? Right now, the story is too positive.

**Questions:**

- How many negative examples are needed before the “opposite-direction” becomes meaningful?

- How many positives (few-shot) do we actually save compared to a strong low-data baseline (e.g., adapter-style fine-tuning or conditional generation with learned property heads)?

- How does the method behave when the true positive set is not just small but diverse (multi-chemotype targets)?
Without a data–performance curve, it’s hard to assess whether this is genuinely more data-efficient or just a different way to consume data.

---

> ### Author Response · Authors · 2025-11-24
> **Response I to Reviewer mQhA**
>
> We thank the reviewer for noting the “clever use of negative data” that clearly addresses an issue that “plagues many molecular design tasks”. Moreover, we agree that model editing is a particularly interesting, unexplored, and practical solution to steer the generation of molecules.
>
> 1. **Task difficulty.** We kindly disagree with the comment. Designing well-docking molecules is a challenging step in bioactive molecule design, where the pretrained models in our study can achieve only 5% success rate. Task arithmetic can double this score via negative data, and achieve over 30% when combined with supervised finetuning on well-docking molecules. In contrast, the suggested distribution learning task in GuacaMol can be saturated even with trivial baselines (doi.org/10.1016/j.ddtec.2020.09.003). We also remark that GuacaMol contains neither scaffold-aware nor isomer/chemotype control tasks.
> In the revised manuscript, we add two more complex design tasks: (i) out-of-distribution generalization, where pretraining has success rates below 5%, and (ii) designing proteins with fewer disordered regions, requiring to capture a complex property. Both analyses show that task arithmetic can design diverse and successful molecules in these settings as well (Section 5.2 and Section 5.6), further supporting our findings.
>
> 2. **Sample efficiency.** We agree that sample efficiency is an important aspect, and the submission manuscript had a subsection reserved for this discussion (Section 5.4). The revised version contains another one, where we discuss out-of-distribution generation capabilities (Section 5.2) in an increasing number of positive and negative molecules.
>
> 3. **More baselines.** None of these baselines is technically comparable with molecular task arithmetic, as they all require positive data/reward function to be available (which is in contrast with the scope of this work). We propose task arithmetic as a training strategy to leverage negative data, and we study whether it can supplement or substitute finetuning for conditional distribution learning. Task arithmetic can even be combined with the proposed alternatives rather than replacing them, e.g., by running an reinforcement learning loop with a task arithmetic prior.
>
> 4. **Failure modes.** The submission already had considerations on validity with two sections (section 5.1 and 5.3) and on diversity, as expressed by the number of clusters (Figs. 2-3, 6-7). Moreover, we disagree with the reviewer on the story being too positive, since we clearly state the limitations of MTA throughout the paper. In the revised version of the manuscript, we have added a specific section on failure modes that collects the main points we already explained in the paper.
>
> ### Answers to Questions
> All of these questions were either fully or partially addressed in Section 5.4 of the submission. The revised version contains another sample efficiency analysis in Section 5.2 (Fig. 3). The short answer is: the 'exchange-rate' between positive and negative molecules depends heavily on the task. While for some design tasks, 10 positive molecules yield better models than task arithmetic on 1024 negative ones, in others (TPSA-related tasks within the OOD setting), over 250 positive molecules are needed to match the performance of 10 negative molecules.

---

### Author Response · Authors · 2025-11-24
**Summary of Edits in the Updated Manuscript (I)**

We thank the reviewers for their comments, which allowed us to remarkably improve our manuscript. We are happy to see a consensus over the significance of the problem we are targeting (low-data molecule design) and the elegance of our approach, molecular task arithmetic (MTA). We have addressed their comments on a point-by-point basis, and carefully updated our manuscript to address their feedback, as follows:

1. **Out-of-distribution generalization (Section 5.2).** To study MTA's performance when there are no positive molecules in the pretraining set, we created six dedicated tasks that require designing molecules whose properties fail outside the training. Our results show that task arithmetic can be effective in such ambitious design settings as well. We also analyzed the performance in increasing sizes of negative and positive molecules for training. This addition targets the points on labeled/unlabeled positive molecules (reviewer ejE6), discussing sample efficiency (reviewer mQhA), and adding another difficult design task (reviewer mQhA).

2. **Additional modalities and model architecture/scales (Section 5.6).** We studied a large-scale pretrained protein language model (ProtGPT2) to design proteins with fewer disordered regions and showed that MTA is applicable also to complex property design tasks (raised by reviewer mQhA) with large-scale transformers and proteins, going beyond LSTMs and small molecules (reviewer ya8s).

3. **Failure modes (Section 5.7)**. We elaborated further on the limitations of MTA in a single subsection (mQhA's and nmQg's comments).

4. **Clarifications.** We further clarified the scope of our work and the metric definitions by considering comments from all reviewers.


While we will keep working on the text to align with ICLR page limits, we hope that with these updates, our work meets the high standards of ICLR.

---

### Author Response · Authors · 2025-11-28
**A Kind Reminder**

Dear reviewers,

We kindly remind you that we are waiting for your responses to shorten the manuscript to ICLR page limits, and to remove highlights on the edited parts.

Authors,
Regards.

---

### Author Response · Authors · 2025-12-03
**Final Remarks**

This work proposed a new transfer learning strategy for de novo drug design of *positive* molecules with *negative* data: molecular task arithmetic (MTA). MTA is a first-of-its-kind approach and uses only negative molecules to learn how to design positive ones. We are grateful to the reviewers for finding MTA an elegant and high-potential approach, as well as for appreciating the clarity of our manuscript. They also suggested new experiments and edits, which we integrated into the manuscript. These new experiments further corroborate the contribution of MTA to tackle the critical lack of positive data in molecular machine learning.

1. We showcased out-of-distribution generalization (suggested by reviewer ejE6; sample efficiency discussion and more difficult design tasks per reviewer mQhA's comments). MTA could increase the number of diverse successful designs from 1K to 15K-50K, even when the training sets contained no positive molecules.

2. We demonstrated the applicability of MTA to other architectures and molecular entities (suggested by reviewers mQhA and ya8s), by designing proteins with fewer disordered regions via large-scale transformers.

3. We aggregated the identified failure modes of MTA into a subsection per mQhA's and nmQg's comments.

4. We re-emphasized MTA's potential to replace and complement supervised fine-tuning, and added preliminary experiments combining it with reinforcement learning (Appendix, Table A6) per suggestion by reviewer mQhA.

Across 33 design settings over molecular entities, model architectures, and task difficulties, MTA showed strong potential to complement, or even replace, current supervised fine-tuning approaches by better balancing diversity and accuracy. Moreover, MTA can be applied to cases where current fine-tuning paradigms are not applicable, i.e., the lack of positive data. The integration of the reviewers' feedback further demonstrates MTA as an elegant, first-of-its-kind approach, with high significance not only for the ICLR community, but for drug discovery in general.

---

### Meta-Review · Area_Chair_UKVf · 2026-01-06

**Summary:**

This is a borderline paper. I mainly checked the reviewers' mQhA and ejE6's comments. Reviewer ejE6's comments are addressed, while reviewer mQhA's concerns are only partially addressed. Plus, that reviewer ya8s's comments are quite modest, and I couldn't tell a strong substantiation. Thus, I tend to reject this paper.

**Reviewer Concerns:**

Reviewer mQhA

- Property space is too easy / narrow. Partially addressed. The reviewer explicitly suggested community-accepted benchmarks (e.g., GuacaMol). The rebuttal dismisses GuacaMol rather than running it, which might leave some reviewers unconvinced.
- No discussion of sample efficiency. Addressed.
- Comparative baselines are underspecified. Not addressed. The answer may not fully satisfy the reviewer, who wants a baseline comparison under the same evaluation protocol, even if the method doesn’t require positives. Saying `cannot compare' may appear evasive unless they add at least a limited comparison (e.g., using a small positive set for RL).
- Limited analysis of failure modes. Addressed.
- Q3 on multi-chemotype. I don't think this is a big issue that should be solved. Thus, this is not considered during evaluation.

---

Reviewer ejE6
- Zero-shot design. Addressed.
- Self-contained. Partially addressed. The reviewer should be more explicit on pointing out which tables/figures should be moved to the main file.
- Cluster definition. This is addressed.

**Reviewer Scores:**

Reviewer ejE6 may update his/her score. Other reviewers may keep the scores.

---

### Decision · Program_Chairs · 2026-01-26

Reject